# Stroke risk in arthritis: A systematic review and meta-analysis of cohort studies

Wei Liu[1☯], Wei Ma[1☯], Hua Liu[2], Chunyan Li[1], Yangwei Zhang[3], Jie Liu[1], Yu Liang[1], Sijia Zhang[1], Zhen Wu[4], Chenghao Zang[4], Jianhui Guo[4]*, Liyan Li[1]*

1 Institute of Neuroscience, Kunming Medical University, Kunming, Yunnan, China, 2 Department of Neurology, The Third People's Hospital of Chengdu & The Affiliated Hospital of Southwest Jiaotong University, Chengdu, Sichuan, China, 3 Department of Neurology, Nanchong Central Hospital & The Second Clinical Medical College, North Sichuan Medical College, Nanchong, Sichuan, China, 4 Second Department of General Surgery, First People's Hospital of Yunnan Province, Kunming, Yunnan, China

☯ These authors contributed equally to this work.
* guojianhuikm@163.com (JG); kmliyanl@163.com (LL)

**Data Availability Statement:** All relevant data are within the paper and its Supporting Information files.

## Abstract

### Background and objective

Stroke is a major contributor to the global burden of disease. Although numerous modifiable risk factors (RF) for stroke have been identified, some remain unexplained. Increasing studies have investigated stroke risk in arthritis, but their results are inconsistent. We aimed to synthesize, quantify, and compare the risk of stroke for the major types of arthritis in cohort studies by using a systematic review and meta-analysis approach.

### Methods

We searched Chinese and English databases to identify relevant studies from inception to April 30, 2020. Only studies adjusting at least for age and sex were included. We calculated pooled effect estimates for relative risk (RR) and 95% confidence interval (CI) and identified potential sources of heterogeneity and publication bias.

### Results

A total of 1,348 articles were retrieved, and after an preliminary screening of titles and abstracts, 69 were reviewed for full text, and finally, 32 met the criteria for meta-analysis. Stroke risk in arthritis was significantly increased in studies adjusting for age and sex (RR = 1.36, 95% CI: 1.27–1.46) and for at least one traditional risk factor (RR = 1.40, 95% CI: 1.28–1.54). The results of studies stratified by stroke subtype were consistent with the main finding (ischemic stroke: RR = 1.53, 95% CI: 1.32–1.78; hemorrhagic stroke: RR = 1.45, 95% CI: 1.15–1.84). In subgroup analysis by arthritis type, stroke risk was significantly increased in rheumatoid arthritis (RR = 1.38, 95% CI: 1.29–1.48), ankylosing spondylitis (RR = 1.49, 95% CI: 1.25–1.77), psoriatic arthritis (RR = 1.33, 95% CI: 1.22–1.45), and gout (RR = 1.40, 95% CI: 1.13–1.73) but not osteoarthritis (RR = 1.03, 95% CI: 0.91–1.16). Age and sex subgroup analyses indicated that stroke risk was similar by sex (women: RR = 1.47, 95% CI: 1.31–1.66; men: RR = 1.44, 95% CI: 1.28–1.61); risk was higher with younger age (<45 years) (RR = 1.46, 95% CI: 1.17–1.82) than older age (≥65 years) (RR = 1.17, 95% CI: 1.08–1.26).

**Funding:** This research was supported by the National Natural Science Foundation of China (No. 31560295, to L.Y.L.). The website of the program is http://www.nsfc.gov.cn/. The funders had no role in study design, data collection and analysis, decision to publish, or preparation of the manuscript.

**Competing interests:** The authors have declared that no competing interests exist.

## Conclusions

Stroke risk was increased in multiple arthritis and similar between ischemic and hemorrhagic stroke. Young patients with arthritis had the highest risk.

## Introduction

Clinically, stroke is a medical condition based on a relatively sudden loss of focal neurological function, broadly categorized as ischemic stroke (IS) or hemorrhagic stroke (HS) [1]. Stroke has become a major public health concern. According to the Global Burden of Disease Study 2016, stroke accounts for almost 5% of all disability-adjusted life-years and 10% of all deaths worldwide, engendering substantial physical and emotional consequences for patients and their families [2]. Thus, there is an urgent need to clearly understand stroke risk. A number of modifiable risk factors have been associated with most of the population attributable risk in stroke worldwide; these include hypertension, diabetes mellitus, obesity, hyperlipidemia, smoking, alcoholism, physical inactivity, diet, psychosocial factors, and cardiac causes [3]. However, studies on these traditional risk factors (RF) cannot fully explain the continuous increase in stroke risk.

There is growing evidence to suggest that inflammation plays a key role in many chronic diseases, including cardiovascular disease, cancer, chronic kidney disease, diabetes, and stroke [4–6]. Arthritis is a chronic inflammatory disease characterized by inflammation of the synovial tissue of the joint [7, 8]. It's inflammation begins with the infiltration of inflammatory cells into the joint [9, 10]. These cells release enzymes, pro-inflammatory factors, and other cytokines that degrade and destroy synovial tissue [9–11], eventually leading to joint pain, deformity, and mobility limitation [12, 13]. The most common types of arthritis are rheumatoid arthritis (RA), osteoarthritis (OA), psoriatic arthritis (PsA), ankylosing spondylitis (AS), gout, and pseudogout [14].

Inflammatory mediators such as cytokines may be the critical factor linking arthritis and stroke. Arthritis involves the increased generation of inflammatory cytokines produced in the joints. These eventually spill into the circulation, where they can cause increased production of adhesion molecules and other proinflammatory molecules. This leads to monocyte and leukocyte adhesion to the endothelial cells of the vessel wall, followed by chemotaxis of these into vessel walls, which leads to atherosclerosis, and ultimately to vascular events such as stroke [15, 16].

Over the past few decades, many epidemiological studies have evaluated the association between arthritis and stroke risk; however, the results have been inconsistent. Wiseman et al. [17] performed a meta-analysis of epidemiological studies on stroke risk in various rheumatic diseases, including RA, AS, and gout. The results showed that the stroke risk of these diseases increased. However, the meta-analysis only included studies up to December 14, 2014. Since then, increasingly more epidemiological studies investigating the stroke risk in numerous types of arthritis have been conducted. However, the consistency and quality of these studies have not been well reviewed, which limits the objective understanding of stroke risk in arthritis.

In epidemiology, cohort studies are preferable to case-control studies and cross-sectional studies for investigating etiological relationships. Generally speaking, age and sex are common potential confounders that might affect the actual result. Adjusting them can focus more on the effect of exposure itself on the outcome. Therefore, we conducted a comprehensive meta-

analysis of cohort studies adjusting at least for age and sex to assess stroke risk in multiple types of arthritis.

## Materials and methods

### Study design

Our review was conducted according to the PRISMA (Preferred Reporting Items for Systematic Reviews and Meta-Analysis) statement [18] and MOOSE (Meta-Analyses and Systematic Reviews of Observational Studies) guidelines [19]. Arthritis was used as an exposure and stroke as an outcome. The present research did not require the approval of an ethics committee and was not registered in any database.

### Data sources and searches

We searched MEDLINE, EMBASE, Cochrane Library, China National Knowledge Infrastructure (CNKI), Weipu, Wanfang, and SINOMed databases using MeSH terms and their entry terms from the initial to April 30, 2020. The example of MeSH terms search strategy in MEDLINE is as follows: ("Stroke"[Mesh]) AND ("Arthritis"[Mesh] OR "Arthritis, Rheumatoid"[Mesh] OR "Arthritis, Psoriatic"[Mesh] OR "Spondylitis, Ankylosing"[Mesh] OR "Gout"[Mesh] OR "Osteoarthritis"[Mesh] OR "Chondrocalcinosis"[Mesh]) AND ("Cohort Studies"[Mesh]). The complete search strategy is presented in S1 Appendix.

All articles searched were stored and managed using Endnote X9 software throughout the review process. We first pooled results from different databases, performed the duplication removal step, and then conducted title and abstract screening followed by full-text screening. In addition, we manually searched the reference lists of identified studies and relevant review articles to identify any studies that were missed in our preliminary search. Articles were not excluded on the basis of language criteria.

### Study selection

Studies that met the following criteria were considered: (i) cohort studies adjusting at least for age and sex; (ii) exposure factors of interest were the six major arthritis types, with clear diagnostic criteria: RA, OA, PsA, AS, gout, and pseudogout; (iii) the outcome of interest was stroke; and (iv) relative risk (RR) and its corresponding 95% CI were reported or the data made available for calculation. Studies that met the following criteria were excluded: (i) not a cohort study, such as a cross-sectional or case-control study; and (ii) the reported RR and 95% CI was not adjusted for age and sex.

If two or more articles were published for the same cohort study, we only included the article with the most detailed information or the longest follow-up period.

### Data extraction

Two investigators used a standard format to extract the following information independently from each study: first author, publication date, country, arthritis type, study design, data source, sample size, study period, age range, proportion of women participants, exposure and outcome validation, stroke subtype, effect estimate type, RR and 95% CI, and additional covariates.

### Quality assessment

We checked the quality of the included studies using the Newcastle-Ottawa Scale (NOS) [20]. NOS is a tool for evaluating the quality of observational studies in systematic reviews and meta-analyses. In the NOS, each study is categorized according to three methodological

characteristics: selection of the study group (scale 0–4); comparability of the study group (scale 0–2); and determination of the exposure or results in case-control or cohort studies (scale 0–3). The highest score for each study is 9. Scores of 0–3, 4–6, and 7–9 are generally regarded as indicative of low, moderate, and high quality, respectively.

### Data synthesis and analysis

RR was used in the statistical analysis of cohort study to measure the strength of the association between the exposure and outcome. Owing to the low absolute risk of stroke, the four correlation indicators were expected to produce similar RR estimates: [standardized mortality rate (SMR), standardized incidence ratio (SIR), incidence rate ratio (IRR), and hazard ratio (HR)] [21]. Therefore, we used these together to assess stroke risk in arthritis, to ensure the comprehensiveness of the evaluation and to maximize statistical power. We used the DerSimonian and Laird random-effects model [22] to calculate the pooled RR and 95% CI for all types of arthritis, first in the studies adjusting only for age and sex, and then in studies adjusting for at least one of the following traditional RF: hypertension, diabetes, smoking, alcoholism, obesity, physical inactivity, and hyperlipidemia. Heterogeneity among studies was assessed using Cochrane's Q statistic (significance level at $p < 0.10$) and the $I^2$ statistic, reflecting the percentage of heterogeneity in the total effect variation. $I^2 < 25\%$ was considered to indicate no heterogeneity and 25%-50% to indicate low, 50%-75% moderate, and $> 75\%$ high heterogeneity [23]. We conducted subgroup analyses to explore the source and magnitude of heterogeneity and examine the effect of different subgroups from a professional perspective. We also performed sensitivity analyses to test the robustness of the results by excluding each study and retesting changes in the size of the combined effect. The publication bias was evaluated using the Begg test and Egger test; $p < 0.10$ was considered statistically significant [24].

All analyses were conducted independently by two authors (WL and WM) using Stata V.14 (StataCorp LLC, College Station, TX, USA). Disagreements were resolved in discussion with a third researcher (HL) until consensus was reached.

## Results

### Literature search and study election

The literature search and screening process are depicted in Fig 1. Our initial search returned 1,339 articles, and an additional 9 articles [25–33] were identified in a manual search. After excluding 144 duplicates, the title and abstract of 1,204 articles were reviewed, of which 1,135 were excluded. After a full-text review of the remaining 69 articles, 37 were omitted for various reasons; a total of 32 articles were finally included [25–56]. The completed PRISMA checklist is given in S1 Checklist.

### Study characteristics and quality

We summarized the main characteristics of the 32 included articles in Table 1. These articles were published from 1994 to 2019, over a span of 25 years. A total of 9 articles were conducted in North America [27, 32, 35, 36, 38, 39, 44, 49, 54], 14 in Europe [28–31, 33, 37, 40–43, 45, 50–52], and 9 in Asia [34, 46–48, 53, 55–58]. There were 10 prospective cohorts (PC) [27, 31, 34, 36, 39, 40, 42, 47, 49, 51] and 22 retrospective cohorts (RC) [28–30, 32, 33, 35, 37, 38, 41, 43–46, 48, 50, 52–58]. Because none of the articles on stroke risk in pseudogout met our inclusion criteria, we eventually only included five types of arthritis (OA, RA, PsA, AS, and gout) in the meta-analysis.

Some articles contained several sub-cohorts; if the researchers only provided the RR and 95% CI for each sub-cohort but not the RR and 95% CI for the total cohort, we treated each

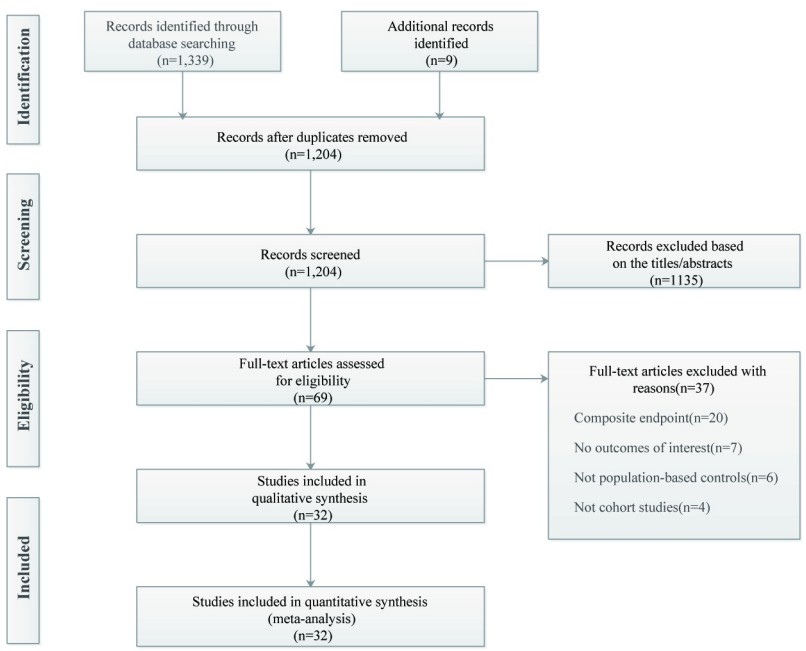

**Fig 1. Flowchart of study selection for the meta-analysis (according to PRISMA statement).**

sub-cohort as an independent cohort study in the meta-analysis. In this way, the 32 articles contained a total of 52 independent cohort studies. Among them, 29 studies were on RA, 4 on OA, 10 on AS, 4 on PsA, 5 on gout, and 31 studies adjusting for at least one traditional RF. The covariates adjusted for each study are shown in S1 Table.

As seen in Table 1, the quality of the 31 included articles was high, with only one article being judged as moderate quality.

## Stroke risk for all arthritis types

Of the 52 independent cohort studies, 31 (59.62%) reported an increased risk of stroke in arthritis. The combined results showed that the stroke risk in arthritis was significantly increased in studies adjusting for age and sex only (RR = 1.36, 95% CI: 1.27–1.46) and for at least one traditional RF (RR = 1.40, 95% CI: 1.28–1.54). A random-effects model was used, owing to the high heterogeneity between studies ($I^2$ = 97%) (Fig 2).

## Stroke risk for each arthritis type

In subgroup analysis according to arthritis type, the results based on studies adjusting for age and sex revealed that stroke risk was significantly increased in RA (RR = 1.38, 95% CI: 1.29–1.48), AS (RR = 1.49, 95% CI: 1.25–1.77), PsA (RR = 1.33, 95% CI: 1.22–1.45), and gout (RR = 1.40, 95% CI: 1.13–1.73); stroke risk was not significantly increased in OA (RR = 1.03, 95% CI: 0.91–1.16). In studies adjusting for at least one traditional RF, the results showed no obvious change compared with those adjusting for age and sex (Fig 3).

## Risk of stroke subtype for all arthritis types

In the 52 studies adjusting for age and sex, 13 separately analyzed the risk of IS, and 6 analyzed the risk of HS; the results suggested that the risk of IS and HS was significantly increased in arthritis (IS: RR = 1.53, 95% CI: 1.32–1.78; HS: RR = 1.45, 95% CI: 1.15–1.84). Of the studies

**Table 1. Characteristics of included studies.**

| Study NO. | First author, Year(ref.no) | Disease | Country | Design | Data source | Study period | Age range | Women % | Arthritis validation | Stroke validation | Exposure total | Effect estimates | NOS quality |
|---|---|---|---|---|---|---|---|---|---|---|---|---|---|
| 1 | Wolfe 1994 [25] | RA | USA&Canada | PC | ARAMIS | 1965–1990 | 53 (mean) | 74.00% | ARA criteria | DC(ID-9) | 3501 | SMR | 9(4/2/3) |
| 2 | Bjornadal 2002 [26] | RA | Sweden | RC | SHDR | 1964–1994 | NA | 71.11% | ICD-7,8,9 | ICD-7,8,9 (DC) | 46917 | SMR | 8(3/2/3) |
| 3 | Solomon 2003 [34] | RA | USA | PC | NHS | 1978–1996 | 30–55 | 100.00% | 1987 ACR | Medical records | NA | IRR | 8(3/2/3) |
| 4 | Watson 2003 [27] | RA | UK | RC | GPRD | 1987–2001 | ≥40 | 69.83% | Medical records | Medical records | 11633 | IRR | 7(3/2/2) |
| 5 | Turesson 2004 [35] | RA | Sweden | RC | RA clinics | 1997–1999 | ≥15 | 73.97% | 1987ACR | ICD-9,10 | 1022 | SMR | 7(4/1/2) |
| 6 | Solomon 2006 [36] | RA | Canada | RC | BCLHD | 1999–2003 | ≥18 | 71.13% | ICD-9 | ICD-9 | 25385 | IRR | 7(3/2/2) |
| 7 | Bergstrom 2009 [28] | RA | Sweden | RC | RA clinic | 1978–1985 | ≥16 | 79.10% | 1958 ARA | ICD-9,10 | 148 | SMR | 7(3/2/2) |
| | | | Sweden | RC | RA clinic | 1995–2002 | ≥16 | 77.60% | 1987 ARA | ICD-9,10 | 161 | SMR | 7(3/2/2) |
| 8 | Semb 2010 [29] | RA | Sweden | PC | AMORIS | 1985–1996 | NA | 68.92% | ICD-8,9,10 | ICD-8,9,10 | 1779 | IRR | 8(4/1/3) |
| 9 | Szabo 2011 [30] | AS | Canada | RC | RAMQ | 1996–2006 | >19 | 43.87% | ICD-9 | ICD-9 | 8616 | SIR | 7(3/1/3) |
| 10 | Brophy 2012 [31] | AS | UK | RC | HER | 1999–2010 | ≥20 | 24.10% | EHR READ | EHR READ | 1686 | IRR | 7(3/1/3) |
| 11 | Li 2012 [37] | PsA | USA | PC | NHS II | 1991–2009 | 25–60 | NA | Self-report | Medical record | NA | IRR | 8(3/2/3) |
| 12 | Lindhardsen 2012 [38] | RA | Denmark | PC | DCVS | 1997–2009 | ≥15 | 69.70% | ICD-10 | ICD-10 | 18247 | IRR | 9(4/2/3) |
| 13 | Teng 2012 [32] | Gout | Singapore | PC | SCHS | 1993–2009 | 49–62 | 34.30% | ICD-9 | ICD-9 | 2117 | HR | 9(4/2/3) |
| 14 | Zoller 2012 [39] | AS | Sweden | RC | NSDR | 1987–2008 | NA | 30.51% | ICD-7,8,9,10 | ICD-9,10 | 3477 | SIR | 7(3/1/3) |
| | | RA | Sweden | RC | NSDR | 1987–2008 | NA | 72.92% | ICD-7,8,9,10 | ICD-9,10 | 44611 | SIR | 7(3/1/3) |
| 15 | Holmqvist 2013 [40] | RA | Sweden | PC | NPR&SPR | 1997–2009 | ≥16 | NA | ICD-10 | ICD-10 | 39065 | HR | 9(4/2/3) |
| 16 | Norton 2013 [41] | RA | UK | RC | ERAS | 1986–1998 | 55.34 (mean) | 66.40% | ICD-10 | ICD-10 | 1460 | SIR | 8(3/2/3) |
| 17 | Rahman 2013 [42] | OA | Canada | RC | BCLHD | 1991–2009 | ≥20 | 60.00% | ICD-9,10 | ICD-9,10 | 12745 | IRR | 7(3/2/2) |
| 18 | Seminog 2013 [43] | Gout | UK | RC | NHS | 1999–2011 | ≥20 | 26.00% | ICD-7,8,9,10 | ICD-10 | 202033 | IRR | 7(3/2/2) |
| | | | UK | RC | ORLS | 1963–1998 | ≥20 | 27.00% | ICD-7,8,9,10 | ICD-10 | 3174 | IRR | 8(3/2/3) |
| 19 | Keller 2014 [44] | AS | China | RC | LHID2000 | 2001 | 42.07 (mean) | 37.80% | ICD-9 | ICD-9 | 2895 | IRR | 7(3/2/2) |
| 20 | Lin 2014 [45] | AS | China | PC | NHI | 2001 | 18–45 | 26.20% | ICD-9 | ICD-9 | 4562 | HR | 8(4/2/2) |
| 21 | Liou 2014 [46] | RA | China | RC | LHID2005 | 2004–2007 | NA | 71.30% | ICD-9 | ICD-9 | 6114 | HR | 7(3/2/2) |
| 22 | Haugen 2015 [47] | OA | USA | PC | FHS | 1990–2015 | 50–75 | 53.80% | Medical records | Medical records | 186 | IRR | 8(3/2/3) |
| 23 | Ogdie 2015 [48] | RA | UK | RC | THIN | 1994–2010 | 18–89 | 70.51% | EHR READ | EHR READ | 41752 | IRR | 9(4/2/3) |

*(Continued)*

**Table 1.** (Continued)

| Study NO. | First author, Year(ref.no) | Disease | Country | Design | Data source | Study period | Age range | Women % | Arthritis validation | Stroke validation | Exposure total | Effect estimates | NOS quality |
|---|---|---|---|---|---|---|---|---|---|---|---|---|---|
| | | PsA | UK | RC | THIN | 1994–2010 | 18–89 | 48.80% | EHR READ | EHR READ | 8706 | IRR | 9(4/2/3) |
| 24 | Bengtsson 2017 [49] | AS | Sweden | PC | SNPR | 2001–2009 | 18–99 | 31.90% | ICD-10 | ICD-10 | 6448 | IRR | 8(4/1/3) |
| | | PsA | Sweden | PC | SNPR | 2001–2009 | 18–99 | 55.10% | ICD-10 | ICD-10 | 16063 | IRR | 8(4/1/3) |
| 25 | Eriksson 2017 [50] | AS | Sweden | RC | NPR&CDR | 2006–2011 | ≥18 | 32.00% | ICD-10 | ICD-10 | 5248 | IRR | 7(3/2/2) |
| | | RA | Sweden | RC | NPR&CDR | 2006–2011 | 63.3 (mean) | 73.00% | ICD-10 | ICD-10 | 35499 | IRR | 7(3/2/2 |
| 26 | Hsu 2017 [51] | OA | China | RC | LHID2000 | 2002–2003 | 20–90 | 59.60% | ICD-9 | ICD-9 | 43635 | IRR | 7(3/1/3) |
| 27 | So 2017 [52] | AS | Canada | RC | BC clinic | 1990–2012 | >18 | 48.70% | ICD-9,10 | ICD-9,10 | 7148 | IRR | 7(3/1/3) |
| 28 | Chen 2018 [53] | RA | China | RC | NHIRD | 2006–2011 | 18–45 | 74.05% | ICD-9 | ICD-9 | 10568 | IRR | 8(3/2/3) |
| 29 | Curtis 2018 [33] | RA | USA | RC | MPCD | 2006–2010 | 40–85 | 80.00% | ICD-9 | ICD-9 | 494 | IRR | 8(3/2/3) |
| 30 | Lee 2018 [54] | AS | South Korea | RC | KNHIS | 2010–2014 | >20 | 27.46% | ICD-10 | ICD-10 | 12988 | IRR | 8(3/2/3) |
| 31 | Tsai 2018 [55] | Gout | China | RC | NHI | 2000–2005 | NA | NA | NA | NA | 646983 | HR | 6(3/1/2) |
| 32 | Kasai 2019 [56] | RA | Janpan | RC | JMDC | 2005–2014 | ≥18 | 75.60% | ICD-10 | ICD-10 | 6712 | IRR | 7(3/2/2) |

ref.no, reference number; ACR, American College of Rheumatology; AMORIS, Apolipoprotein Mortality RISk; ARA, American Rheumatism Association; ARAMIS, Arthritis, Rheumatism, and Aging Medical Information System; AS, ankylosing spondylitis; BC, British Columbia; BCLHD, British Columbia Longitudinal Health Database; DCVS, Danish civil registration system; ERAS, Early RA Study; FHS, Framingham Heart Study; GPRD, General Practice Research Database; HER, electronic health record; JMDC, Japan Medical Data Center; KNHIS, Korean NationalHealthInsurance Service; LHID, Longitudinal Health Insurance Database; MPCD, multi-payer claims database; NHI, National Health Insurance; NHIRD, National Health Insurance Research Database; NHS, National Health Service; NOS, Newcastle-Ottawa Scale; NPR&SPR, The National Patient Register and the Swedish Population Register; NSDR, National Swedish data registers; OA, osteoarthritis; ORLS, Oxford Record Linkage Study; PC, prospective cohort study; PsA, psoriatic arthritis; RA, Rheumatoid arthritis; RAMQ, Re´gie de l'Assurance Maladie du Que´bec database; RC, retrospective cohort study; SCHS, Singapore Chinese Health Study; SHDR, Swedish Hospital Discharge Register; SNPR, Swedish National Patient Register; THIN, The Health Improvement Network; TLHID, Taiwan's Longitudinal Health Insurance Database.

adjusting for at least one traditional RF, the risk of IS and HS was also significantly increased (IS: RR = 1.59, 95% CI: 1.32–1.92; HS: RR = 1.63, 95% CI: 1.23–2.15) (Table 2).

## Age and sex subgroup analyses

In subgroup analysis based on sex, the risk was similar (women: RR = 1.47, 95% CI: 1.31–1.66; men: RR = 1.44, 95% CI: 1.28–1.61). Subgroup analysis stratified by age revealed that stroke risk was highest with younger age (<45 years) (RR = 1.46, 95% CI: 1.17–1.82) and relatively lower with older age (≥65 years) (RR = 1.17, 95% CI: 1.08–1.26) (Table 2).

## Region and cohort type subgroup analyses

In the region-based subgroup analysis, the stroke risk was similar in Asia and Europe but slightly higher in North America (Asia: RR = 1.26,95%CI:1.18–1.35; Europe: RR = 1.39,95% CI:1.31–1.48; North America: RR = 1.28,95%CI:1.06–1.55). Subgroup analysis by cohort study

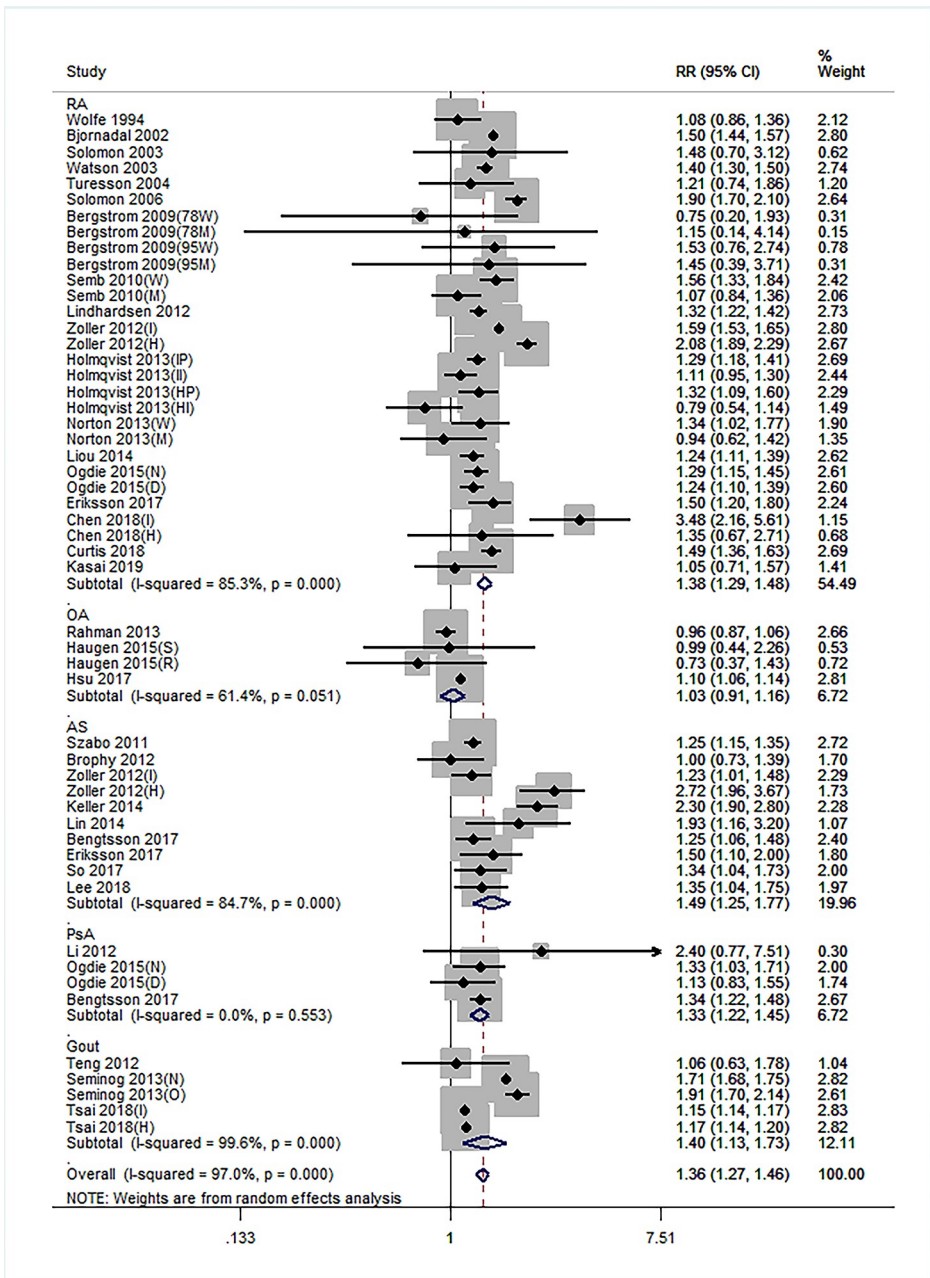

**Fig 2. Forest plot showing the stroke risk in arthritis in studies adjusted for age and sex.**

type showed that RC studies was slightly higher than PC studies (RC: RR = 1.42,95%CI:1.31–1.53; PC: RR = 1.26,95%CI:1.17–1.35) (Table 2).

## Heterogeneity testing and sensitivity analysis

The combined analysis of 52 studies showed high heterogeneity ($I^2$ = 97%, $p < 0.001$). To explore the source of heterogeneity, we first performed subgroup analyses by arthritis type, stroke subtype, sex, age, effect estimate type, region, cohort study type, publication date. However, the results showed that they all could not explain the source of heterogeneity (Table 2).

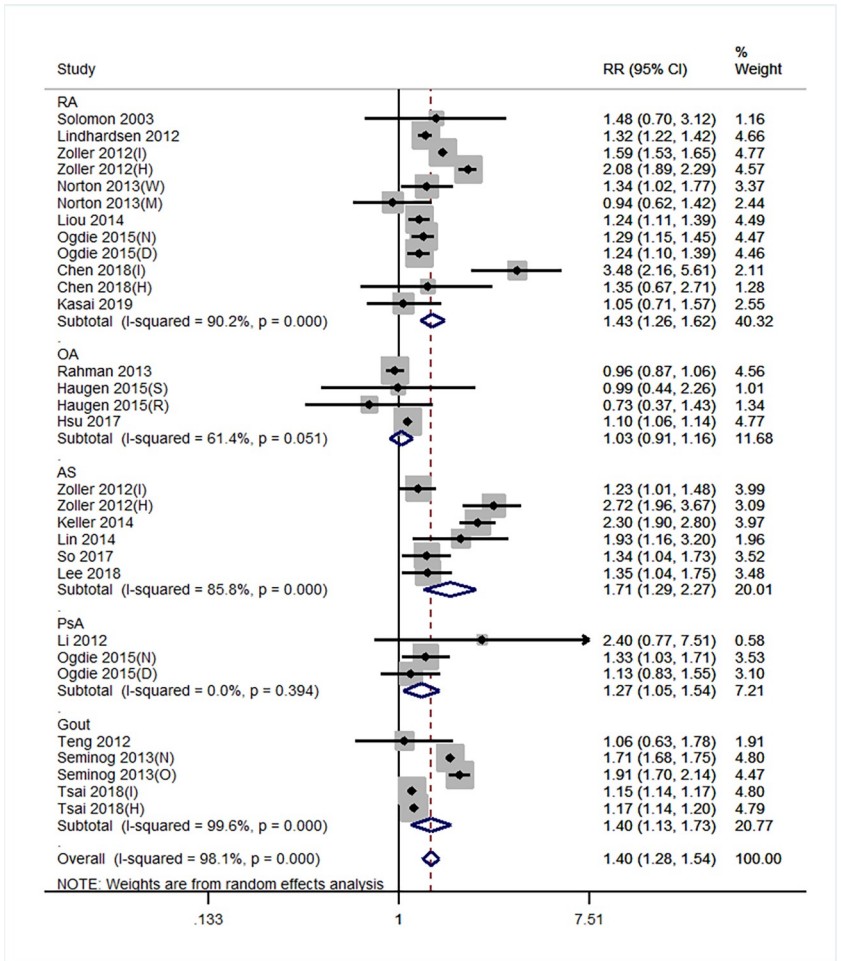

**Fig 3. Forest plot showing the stroke risk in arthritis in studies adjusted for age and sex and at least one traditional risk factor (hypertension, diabetes, smoking, alcoholism, obesity, physical inactivity, and hyperlipidemia).**

We further performed sensitivity analyses to investigate the source of significant heterogeneity and to test the impact of each single study on the final results. We found that no single study qualitatively changed the pooled effects, indicating the reliability and stability of our results. We also found that no single study had a significant effect on heterogeneity; with the $I^2$ statistic reduced to < 50%, 14 studies had to be excluded.

## Publication bias

The Egger test ($p = 0.435$) and Begg test ($p = 0.313$) were not statistically significant, and the funnel plot was roughly symmetrical, indicating that no significant publication bias was present (Fig 4).

## Discussion

In the past few decades, numerous studies on risk factors for stroke have focused on traditional RF such as hypertension, diabetes, obesity, smoking, alcoholism, and physical inactivity. There has been a growing number of studies on new RF for stroke in recent years. However, the

**Table 2. Subgroup analyses of population-based studies estimating stroke risk in the major types of arthritis.**

| Comparison | Studies adjusting for age and sex only | | | Studies adjusting for age, sex and at least one traditional RF[#] | | |
|---|---|---|---|---|---|---|
| | Study number | Random-effects RR (95% CI) | *P* value | Study number | Random-effects RR (95% CI) | *P* value |
| **Stroke Type** | | | | | | |
| IS | 14 | 1.53(1.32, 1.78) | <0.001 | 10 | 1.59(1.32, 1.92) | <0.001 |
| HS | 7 | 1.45(1.15, 1.84) | <0.001 | 5 | 1.63(1.23, 2.15) | <0.001 |
| **Sex** | | | | | | |
| Female | 20 | 1.47(1.31, 1.66) | <0.001 | 11 | 1.68(1.43, 1.97) | <0.001 |
| Male | 20 | 1.44(1.28, 1.61) | <0.001 | 11 | 1.53(1.33, 1.77) | <0.001 |
| **Age** | | | | | | |
| <45 | 4 | 1.46(1.17, 1.82) | 0.001 | 2 | 1.60(0.70, 3.62) | 0.262* |
| 45–64 | 11 | 1.43(1.18, 1.72) | <0.001 | 5 | 1.57(1.19, 2.07) | 0.001 |
| ≥65 | 10 | 1.17(1.08, 1.26) | <0.001 | 4 | 1.12(1.04, 1.21) | 0.004 |
| **Cohort Type** | | | | | | |
| PC | 16 | 1.26(1.17, 1.35) | <0.001 | 7 | 1.30(1.08, 1.57) | 0.005 |
| RC | 36 | 1.42(1.31, 1.53) | <0.001 | 23 | 1.42(1.28, 1.57) | <0.001 |
| **Region** | | | | | | |
| Asia | 11 | 1.26(1.18, 1.35) | <0.001 | 11 | 1.26(1.18, 1.35) | <0.001 |
| Europe | 31 | 1.39(1.31, 1.48) | <0.001 | 13 | 1.50(1.38, 1.64) | <0.001 |
| North America | 10 | 1.28(1.06, 1.55) | 0.011 | 6 | 1.11(0.88, 1.41) | 0.386* |
| **Publication Date** | | | | | | |
| ≤2009 | 10 | 1.46(1.29, 1.64) | <0.001 | 1 | 1.48(0.70, 3.12) | 0.304* |
| ≥2010 | 42 | 1.35(1.26, 1.46) | <0.001 | 29 | 1.40(1.28, 1.54) | <0.001 |
| **Effect Estimate** | | | | | | |
| IRR | 29 | 1.39(1.26, 1.54) | <0.001 | 19 | 1.39(1.20, 1.60) | <0.001 |
| SIR | 7 | 1.53(1.28, 1.81) | <0.001 | 6 | 1.59(1.31, 1.93) | <0.001 |
| HR | 9 | 1.18(1.14, 1.23) | <0.001 | 5 | 1.16(1.13, 1.20) | <0.001 |
| SMR | 7 | 1.31(1.09, 1.57) | 0.004 | NA | NA | NA |

RR, relative risk; PC, prospective cohort study; RC, retrospective cohort study; SMR, standardized mortality rate; SIR, standardized incidence ratio; IRR, incidence rate ratio; HR, hazard ratio; RF, risk factor; NA, not applicable.

* No statistical significance.

[#]traditional RF: hyperlipidemia, diabetes, high blood pressure, smoking, obesity and physical activity.

sample size in most studies was small and the conclusions were inconsistent. Therefore, some researchers merged multiple studies in quantitative meta-analyses and found a series of risk factors involved in stroke, including migraine, anemia, inflammatory bowel disease, sleep insufficiency, insufficient intake of fruits and vegetables, and inflammatory diseases [57–61]. Wiseman et al. [17] conducted a meta-analysis of studies on stroke risk in rheumatic diseases and found that RA, AS, gout, and systemic lupus erythematosus increased stroke risk to varying degree. Since these studies, there have been no meta-analyses of stroke risk in any arthritis type, which may lead to underestimation of the actual stroke risk in clinical practice.

To date, our review is the most comprehensive systematic review and meta-analysis of published cohort studies to evaluate the stroke risk in arthritis. We compared the stroke risk in multiple types of arthritis, not only adjusting for age and sex but also adjusting for traditional RF, which may be confounding factors for arthritis. Our review suggested that individuals with arthritis had a 36% higher risk of developing stroke than the general population. This relationship was also found in stroke subgroup analysis, with a 53% higher risk of IS and a 45% higher risk of HS. Compared with middle-aged and older patients, younger patients had the

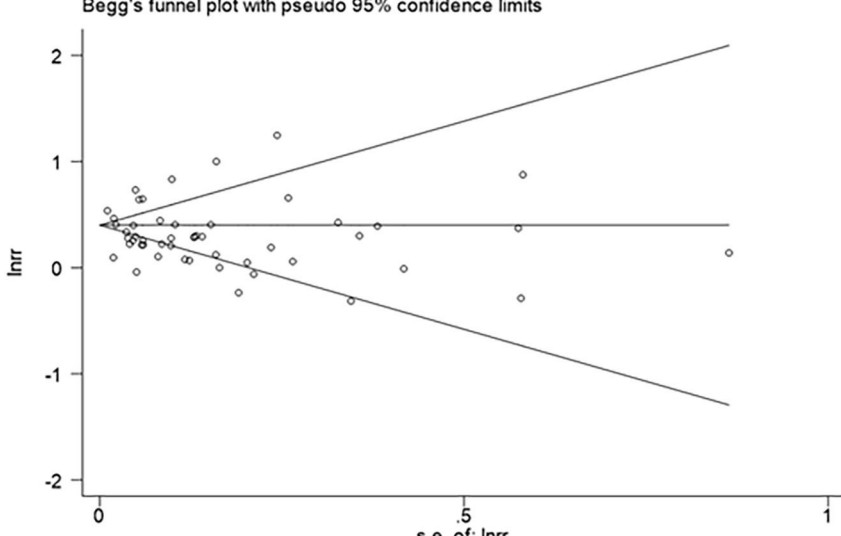

**Fig 4. Funnel plot for stroke risk in arthritis.**

highest stroke risk. Interestingly, a few years ago, Fransen et al. [62] performed a similar meta-analysis in which they found a higher CVD risk among younger RA patients. As young people ordinarily have fewer traditional RF, we speculate that arthritis is an independent risk factor that does not depend on traditional RF. Besides, In 2017, Schieir et al. [63] conducted a meta-analysis on the risk of myocardial infarction (MI) in arthritis. They found that the MI risk in all types of arthritis was attenuated in studies adjusting for traditional RF, in comparison with unadjusting studies, further suggesting that traditional RF could partially explain the MI risk in arthritis. Unlike their conclusion, our review showed that traditional RF could not explain the stroke risk in arthritis because this risk adjusting for traditional RF was slightly higher than the stroke risk in arthritis without adjustment for traditional RF. Taken together, we have good reasons to consider arthritis as an independent risk factor for stroke.

Owing to the rising incidence rate of arthritis, our review is of great clinical importance [64]. First, we assessed the implications of the review from the perspective of patients. Stroke risk in RA has been widely studied and is generally widely recognized [65], whereas stroke risk in other types of arthritis has received much less attention and may not be fully recognized, especially in OA and PsA. We found an increased risk of stroke in the most common types of arthritis; therefore, stroke can be considered a common complication of arthritis and should be taken more seriously. Second, OA is generally considered non-systemic inflammatory arthritis; the other four types of arthritis (RA, AS, PsA, gout) are considered systemic inflammatory arthritis. Our review showed that OA has no additional risk of stroke, suggesting that systemic inflammation may be the direct cause of increased risk of stroke. From the clinician's perspective, this information is essential because clinicians can prioritize the control of inflammation among traditional RF, which may help to reduce the stroke risk. Lastly, the association between arthritis and stroke is important from a public health perspective. It is necessary to consider screening arthritis status and RF for cardio-cerebrovascular diseases in the general population for early intervention to reduce future stroke events.

The strengths of our review include the following: (i) because of cohort studies' powerful ability to test the RF hypothesis, we only included cohort studies in our meta-analysis, rendering our conclusions relatively more reliable. (ii) compared with a study by Wiseman et al., we

conducted a more comprehensive search and had a much larger sample size and broader study area, compensating for the lack of information on Asian patients with arthritis in that previous analysis. (iii) for the first time, we compared the stroke risk for multiple types of arthritis in studies adjusting for age and sex, which are the most common confounding factors, then studies adjusting for traditional RF, which helps in determining whether arthritis is a RF independent of traditional RF.

This review includes several limitations: (i) there was a high degree of heterogeneity between studies. Although we conducted subgroup analyses and sensitivity analyses, we were unable to identify potential sources of heterogeneity. We assumed that the statistical heterogeneity was primarily attributable to the degree of variability in effect size across different studies; therefore, we used a random-effects model to explain variability across studies. (ii) although we have excluded explicitly duplicated studies, there may be some overlap among the populations of more than one study from the same cohort. (iii) 69% of the included studies were retrospective, this may lead to biases inherent in these types of studies.

More than 60% of the included studies did not explore stroke subtypes, more than 50% did not carry out age stratification, and more than 40% did not adjust for any traditional RF. Therefore, additional large-sample cohort studies are needed to explore the effects of different types of arthritis on stroke and other vascular end-point events; such studies should be stratified by stroke subtype, age, and sex and should adjust for traditional RF. In addition, further clinical studies are also needed to identify improved treatments for different types of arthritis.

## Conclusions

Although observational studies cannot prove causality, the findings of this meta-analysis of large, high-quality population-based cohort studies strongly support that multiple types of arthritis increase stroke risk. These findings provide additional reliable evidence for arthritis as an independent RF for stroke, which requires greater attention among patients with arthritis. However, considering the potential bias and confounding in the included studies, caution should be exercised in the interpretation of the results.

## Supporting information

**S1 Checklist. PRISMA 2009 checklist used in this meta-analysis.**
(DOCX)

**S1 Appendix. Sample MEDLINE search strategy.**
(DOCX)

**S1 Table. The covariates for adjustment in each study.**
(DOCX)

## Acknowledgments

We would like to thank Dr. Yingchuan Zhu (West China Medical School, Sichuan University) for her kindly help with the final version of the paper.

## Author Contributions

**Data curation:** Wei Liu.

**Formal analysis:** Wei Liu, Wei Ma, Hua Liu, Yu Liang, Sijia Zhang, Jianhui Guo.

**Funding acquisition:** Liyan Li.

**Investigation:** Wei Liu, Chunyan Li, Jie Liu.

**Methodology:** Wei Liu, Wei Ma, Hua Liu, Chunyan Li, Zhen Wu, Chenghao Zang, Liyan Li.

**Project administration:** Wei Liu, Wei Ma, Liyan Li.

**Resources:** Jianhui Guo.

**Software:** Yangwei Zhang.

**Supervision:** Jianhui Guo, Liyan Li.

**Writing – original draft:** Wei Liu, Wei Ma.

**Writing – review & editing:** Jianhui Guo, Liyan Li.

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
