## [Decision Letter · Decision Letter 0]

15 Jan 2021

PONE-D-20-19285

Stroke risk in arthritis: A systematic review and meta-analysis of vohort studies

PLOS ONE

Dear Dr. Li,

Thank you for submitting your manuscript to PLOS ONE. After careful consideration, we feel that it has merit but does not fully meet PLOS ONE’s publication criteria as it currently stands. Therefore, we invite you to submit a revised version of the manuscript that addresses the points raised during the review process.

The title needs to be revised as well: "cohort" not "vohort".

We look forward to receiving your revised manuscript.

Kind regards,

Yiqiang Zhan

Academic Editor

PLOS ONE

Reviewers' comments:

Reviewer's Responses to Questions

**Comments to the Author**

1. Is the manuscript technically sound, and do the data support the conclusions?

Reviewer #1: Yes

Reviewer #2: Yes

2. Has the statistical analysis been performed appropriately and rigorously? 

Reviewer #1: Yes

Reviewer #2: I Don't Know

3. Have the authors made all data underlying the findings in their manuscript fully available?

Reviewer #1: Yes

Reviewer #2: Yes

4. Is the manuscript presented in an intelligible fashion and written in standard English?

Reviewer #1: Yes

Reviewer #2: Yes

5. Review Comments to the Author

Reviewer #1: Thank you for the opportunity to review this research article – Stroke risk in arthritis: A systematic review and meta-analysis of cohort studies . Please find my comments below:

Overall comments: This systematic review and meta-analysis would be a valuable addition to the literature around risk of stroke in arthritis. I would like to compliment authors for maintaining rigour in the conduct of this review and for providing adequate details to convey their findings. However, I would like to make few constructive comments to add more clarity and further strengthen this manuscript. Please see my specific comments below:

Specific comments:

Title: Please correct the word “Vohort” to “Cohort” in the title.

Abstract: The abstract is well written and concise enough to provide readers with the snapshot of the study. I would like to make a minor comment. I would suggest to authors to include number of articles screened for titles and abstracts, then for full text and finally mention the total number of included studies (32). This would help readers to understand the breadth of the review.

Introduction: Authors have emphasised heavily on including studies that adjusted for age and sex. However, I did not find any explanation around the rationale for such emphasis. Adding information on how age and sex could have impact on stroke and arthritis would lay a stronger foundation for the manuscript. Considering the broad readership of PLOS One Journal, I would suggest adding biological mechanism behind the potential relationship between arthritis and stroke.

Material and Methods:

Study design: It would have been ideal to register this review on one of the databases such as, PROSPERO. But it is late to do so, I would not advice doing it now. I would suggest authors be cognizant of registering their future systematic reviews.

Authors have jumped directly from study design to study selection. Please report the steps from screening process. Was title and abstract screening done in duplicates, was any reference manager used ? Was there any language criteria used to exclude studies ?

Results:

In Table 1, please add first column as study number and add reference for each study in the table (for readers to easily access reference of each study in the table)

Data sources and searches:

Authors have very comprehensive search strategy, but most of the readers do not read appendices, so I would suggest adding few keywords from search strategy in the manuscript.

Discussion and Conclusion:

These sections are very well written.

Reviewer #2: A good manuscript providing important information related to the problem of stroke in patients with various inflammatory rheumatic conditions. The authors found higher risks for both ischemic as well as hemorrhagic stroke, persisting after adjustments were done for age and gender. I have a few issues which I would like to be addressed by the authors.

1. MEthods/Results: the authors report 32 studies elligible for their analyses. Yet, they also mention that they have included 52 studies (Results, page 9). Can the authors clarify this apparent discrepancey? Thank you.

2. The authors included studies from cohorts of patients situated on different continents. Did they peformed a subanalysis for every continent (Europe, North America and Asia, respectively) and did they obtained the same/ similar results (higher risk of stroke)? This information is interesting for the reader as it might uncover the possible contribution of genetic factors.

3. The authors mention that 69% of the studies were retrospective, meaning that 31% were prospective studies. Was there a difference in the risk of stroke between the data pooled for each of this two kind of cohorts? If not, some biases could be looked with less caution.

4. An interesting finding of the present study is the higher risk of stroke among younger patients. Few years ago, Fransen et al (Fransen J et al, PloS One; 2016; 11:e0157360) performed a similar meta-analysis in which they found a higher cardiovascular risk among younger RA patients. Can the authors elude in their discussion how this previous findings for RA could be integrated with their own findings and speculate on the pathofysiological mechanisms? This should be of interest for the readers.

6. PLOS authors have the option to publish the peer review history of their article (what does this mean?). If published, this will include your full peer review and any attached files.

Reviewer #1: No

Reviewer #2: No

---

## [Author Response · Author response to Decision Letter 0]

26 Feb 2021

Dear Prof. Yiqiang Zhan,

Academic Editor

PLOS ONE

Thank you very much for your letter and advice. We have carefully revised the manuscript according to the comments and suggestions raised by reviewers. We would like to re-submit it for your consideration. We provided our point-by-point responses to the comments and suggestions below. The revised parts are marked in red in the revised manuscript.

The word “vohort” has been corrected to “cohort” in the title.

We hope that the revision is acceptable, and I look forward to hearing from you soon.

Best wishes,

Sincerely,

Liyan Li PhD

E-mail: kmliyanl@163.com

Responses to reviewer comments

Reviewer #1 comments:

Comment 1. Title: Please correct the word “Vohort” to “Cohort” in the title.

Response: We have corrected the word "vohort" to "cohort" in the title.

Comment 2. Abstract: The abstract is well written and concise enough to provide readers with the snapshot of the study. I would like to make a minor comment. I would suggest to authors to include number of articles screened for titles and abstracts, then for full text and finally mention the total number of included studies (32). This would help readers to understand the breadth of the review.

Response: Your suggestions are greatly appreciated. We have added the description of articles screening that included the number of titles, abstracts, full text, and final included studies as follows:

“A total of 1,348 articles were retrieved, and after a preliminary screening of titles and abstracts, 69 were reviewed for full text, and finally, 32 met the criteria for meta-analysis.”

Comment 3. Introduction: Authors have emphasised heavily on including studies that adjusted for age and sex. However, I did not find any explanation around the rationale for such emphasis. Adding information on how age and sex could have impact on stroke and arthritis would lay a stronger foundation for the manuscript. Considering the broad readership of PLOS One Journal, I would suggest adding biological mechanisms behind the potential relationship between arthritis and stroke.

Response: Thanks for your comments and suggestions. In epidemiological studies, age and sex are common potential confounding factors that might affect the actual result. Adjusting them can focus more on the effect of exposure itself on the outcome, so we chose studies adjusting for at least age and sex for our meta-analysis. The explanation for this has been added in the revised file (page 5-6, line102-107). With regard to the biological mechanisms behind the potential relationship between arthritis and stroke, we have added this part in the revised file (page 4-5, line77-92) after reviewing the extensive literature.

Comment 3. Study design: It would have been ideal to register this review on one of the databases, such as PROSPERO. But it is late to do so, I would not advice doing it now. I would suggest authors be cognizant of registering their future systematic reviews. Authors have jumped directly from study design to study selection. Please report the steps from screening process. Was title and abstract screening done in duplicates, was any reference manager used ? Was there any language criteria used to exclude studies ?

Response: Thank you for your instructive suggestions. We will register our future systematic reviews on one of the databases, such as PROSPERO. We have added a detailed description of the steps from literature screening process in the “Data sources and searches” section as follows: 

“All articles searched were stored and managed using Endnote X9 software throughout the review process. We first pooled results from different databases, performed the duplication removal step, and then conducted title and abstract screening followed by full-text screening. In addition, we manually searched the reference lists of identified studies and relevant review articles to identify any studies that were missed in our preliminary search. Articles were not excluded on the basis of language criteria.”

Comment 4. Results: In Table 1, please add first column as study number and add reference for each study in the table (for readers to easily access reference of each study in the table).

Response: We have added the first column as the study number in Table 1 and modified the second column's name to "First author, Year(ref.no)," and then added the reference number of each study.

Comment 5. Data sources and searches: Authors have very comprehensive search strategy, but most of the readers do not read appendices, so I would suggest adding few keywords from search strategy in the manuscript.

Response: We used MeSH terms and their entry terms for searching. We have added the MeSH terms search strategy in MEDLINE as examples in the revised manuscript to give readers a general understanding of our search strategy.

“The example of MeSH terms search strategy in MEDLINE is as follows: ("Stroke"[Mesh]) AND ("Arthritis"[Mesh] OR "Arthritis, Rheumatoid"[Mesh] OR "Arthritis, Psoriatic"[Mesh] OR "Spondylitis, Ankylosing"[Mesh] OR "Gout"[Mesh] OR "Osteoarthritis"[Mesh] OR "Chondrocalcinosis"[Mesh]) AND ("Cohort Studies"[Mesh]).”

Reviewer #2 comments:

Comment 1. Methods/Results: the authors report 32 studies elligible for their analyses. Yet, they also mention that they have included 52 studies (Results, page 9). Can the authors clarify this apparent discrepancey? Thank you.

Response: Your suggestions are greatly appreciated. 32 is actually the number of articles included, 52 is the number of independent cohort studies. We have made a revision to make a clear distinction between the "articles" and the "studies" as follows:

“Some articles contained several sub-cohorts; if the researchers only provided the RR and 95% CI for each sub-cohort but not the RR and 95% CI for the total cohort, we treated each sub-cohort as an independent cohort study in the meta-analysis. In this way, the 32 articles contained a total of 52 independent cohort studies.”

Comment 2. The authors included studies from cohorts of patients situated on different continents. Did they peformed a subanalysis for every continent (Europe, North America and Asia, respectively) and did they obtained the same/ similar results (higher risk of stroke)? This information is interesting for the reader as it might uncover the possible contribution of genetic factors.

Response: We performed a subgroup analysis by region (Europe, North America, and Asia). The results were not elaborated in the text but were shown in Table 2. According to your suggestion, we have added this part on page 16. Since the cohort studies in North America and Europe only included pooled RR(95%CI) for multiple ethnicities(Caucasians, Asians, and Africans), yet did not have RR(95%CI) for each ethnicity, we could not perform a subgroup analysis by ethnicity and observe the possible contribution of genetic factors.

Comment 3. The authors mention that 69% of the studies were retrospective, meaning that 31% were prospective studies. Was there a difference in the risk of stroke between the data pooled for each of this two kind of cohorts? If not, some biases could be looked with less caution.

Response: We also conducted a subgroup analysis by cohort study type [retrospective cohort (RC) and prospective cohort (PC)]. The results were not elaborated in the text but were shown in Table 2. We have added this part on page 16. It is well known that compared with PC, the bias of RC is relatively greater, which may overestimate the real results. We assume that this may be the main reason why the RR of RC is slightly higher than PC in this study. Therefore, our results should still be viewed with some caution.

Comment 4. An interesting finding of the present study is the higher risk of stroke among younger patients. Few years ago, Fransen et al (Fransen J et al, PLOS ONE; 2016; 11:e0157360) performed a similar meta-analysis in which they found a higher cardiovascular risk among younger RA patients. Can the authors elude in their discussion how this previous findings for RA could be integrated with their own findings and speculate on the pathofysiological mechanisms? This should be of interest for the readers.

Response: Thank you for your advice. We carefully read the meta-analysis by Fransen et al. The conclusion is similar to ours. We have explored the possible pathophysiological mechanisms by an extensive literature search in MEDLINE and other databases. Regrettably, we did not retrieve any published hypothesis, so we could not speculate its pathophysiological mechanisms currently.

---

## [Editor Report · Decision Letter 1]

2 Mar 2021

Stroke risk in arthritis: A systematic review and meta-analysis of cohort studies

PONE-D-20-19285R1

Dear Dr. Li,

We’re pleased to inform you that your manuscript has been judged scientifically suitable for publication and will be formally accepted for publication once it meets all outstanding technical requirements.

Kind regards,

Y Zhan

Academic Editor

PLOS ONE
---

## [Editor Report · Acceptance letter]

4 Mar 2021

PONE-D-20-19285R1 

Stroke Risk in Arthritis: a Systematic Review and Meta-analysis of Cohort Studies 

Dear Dr. Li:

I'm pleased to inform you that your manuscript has been deemed suitable for publication in PLOS ONE. Congratulations! Your manuscript is now with our production department. 

Kind regards, 

on behalf of

Dr. Y Zhan 

Academic Editor

PLOS ONE